# LARGE-SCALE STUDY OF CURIOSITY-DRIVEN LEARNING

## ABSTRACT

Reinforcement learning algorithms rely on carefully engineered rewards from the environment that are extrinsic to the agent. However, annotating each environment with hand-designed, dense rewards is difficult and not scalable, motivating the need for developing reward functions that are intrinsic to the agent. Curiosity is such intrinsic reward function which uses prediction error as a reward signal. In this paper: (a) We perform the first large-scale study of purely curiosity-driven learning, i.e. *without any extrinsic rewards*, across $54$ standard benchmark environments, including the Atari game suite. Our results show surprisingly good performance as well as a high degree of alignment between the intrinsic curiosity objective and the hand-designed extrinsic rewards of many games. (b) We investigate the effect of using different feature spaces for computing prediction error and show that random features are sufficient for many popular RL game benchmarks, but learned features appear to generalize better (e.g. to novel game levels in Super Mario Bros.). (c) We demonstrate limitations of the prediction-based rewards in stochastic setups. Game-play videos and code are at https://doubleblindsupplementary.github.io/large-curiosity/.

## 1 INTRODUCTION

Reinforcement learning (RL) has emerged as a popular method for training agents to perform complex tasks. In RL, the agent's policy is trained by maximizing a reward function that is designed to align with the task. The rewards are extrinsic to the agent and specific to the environment they are defined for. Most of the success in RL has been achieved when this reward function is dense and well-shaped, e.g., a running "score" in a video game (Mnih et al., 2015). However, designing a well-shaped reward function is a notoriously challenging engineering problem. An alternative to "shaping" an extrinsic reward is to supplement it with dense intrinsic rewards (Oudeyer & Kaplan, 2009), that is, rewards that are generated by the agent itself. Examples of intrinsic reward include "curiosity" (Mohamed & Rezende, 2015; Schmidhuber, 1991b; Singh et al., 2005; Houthooft et al., 2016; Pathak et al., 2017) which uses prediction error as reward signal, and "visitation counts" (Bellemare et al., 2016; Ostrovski et al., 2017; Poupart et al., 2006; Lopes et al., 2012) which discourages the agent from revisiting the same states. The idea is that these intrinsic rewards will bridge the gaps between sparse extrinsic rewards by guiding the agent to efficiently explore the environment to find the next extrinsic reward.

But what about scenarios with no extrinsic reward at all? This is not as strange as it sounds. Developmental psychologists talk about *intrinsic motivation* (i.e., curiosity) as the primary driver in the early stages of development (Smith & Gasser, 2005; Ryan, 2000): babies appear to employ goal-less exploration to learn skills that will be useful later on in life. There are plenty of other examples, from playing Minecraft to visiting your local zoo, where no extrinsic rewards are required. Indeed, there is evidence that pre-training an agent on a given environment using only intrinsic rewards allows it to learn much faster when fine-tuned to a novel task in a novel environment (Pathak et al., 2017; 2018). Yet, so far, there has been no systematic study of learning with only intrinsic rewards.

In this paper, we perform a large-scale empirical study of agents driven purely by intrinsic rewards across a range of diverse simulated environments. In particular, we choose the dynamics-based curiosity model of intrinsic reward presented in Pathak et al. (2017) because it is scalable and trivially parallelizable, making it ideal for large-scale experimentation. The central idea is to represent intrinsic reward as the error in predicting the consequence of the agent's action given its current state,

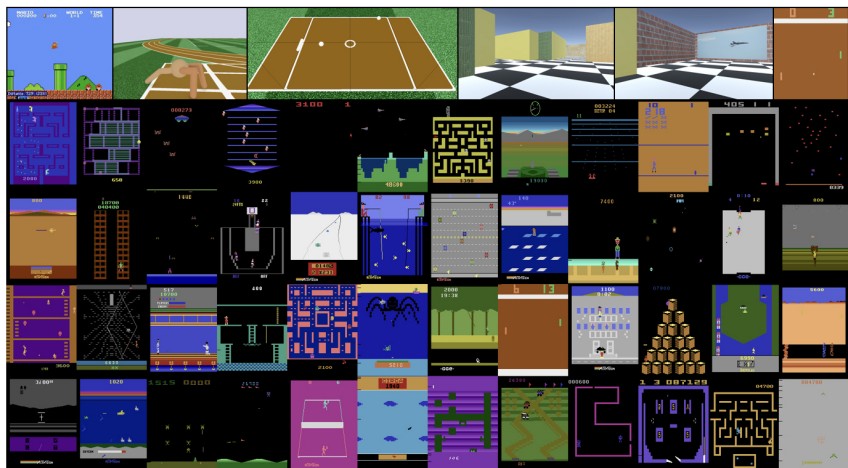

Figure 1: A snapshot of the 54 environments investigated in the paper. We show that agents are able to make progress using no extrinsic reward, or end-of-episode signal, and only using curiosity. Video results, code and models at https://doubleblindsupplementary.github.io/large-curiosity/.

i.e., the prediction error of learned forward-dynamics of the agent. We thoroughly investigate the dynamics-based curiosity across 54 environments: video games, physics engine simulations, and virtual 3D navigation tasks, shown in Figure 1.

To develop a better understanding of curiosity-driven learning, we further study the crucial factors that determine its performance. In particular, predicting the future state in the high dimensional raw observation space (e.g., images) is a challenging problem and, as shown by recent works (Pathak et al., 2017; Stadie et al., 2015), learning dynamics in an auxiliary feature space leads to improved results. However, how one chooses such an embedding space is a critical, yet open research problem. To ensure stable online training of dynamics, we argue that the desired embedding space should: 1) be *compact* in terms of dimensionality, 2) preserve *sufficient* information about the observation, and 3) be a *stationary* function of the observations.

Through systematic ablation, we examine the role of different ways to encode agent's observation such that an agent can perform well, driven purely by its own curiosity. Here "performing well" means acting purposefully and skillfully in the environment. This can be assessed quantitatively, in some cases, by measuring extrinsic rewards or environment-specific measures of exploration, or qualitatively, by observing videos of the agent interacting. We show that encoding observations via a random network turn out to be a simple, yet surprisingly effective technique for modeling curiosity across many popular RL benchmarks. This might suggest that many popular RL video game test-beds are not as visually sophisticated as commonly thought. Interestingly, we discover that although random features are sufficient for good performance in environments that were used for training, the learned features appear to generalize better (e.g., to novel game levels in Super Mario Bros.).

The main contributions of this paper are: (a) Large-scale study of curiosity-driven exploration across a variety of environments including: the set of Atari games (Bellemare et al., 2013), Super Mario Bros., virtual 3D navigation in Unity (Juliani et al., 2018), multi-player Pong, and Roboschool (Schulman et al., 2017) environments. (b) Extensive investigation of different feature spaces for learning the dynamics-based curiosity: random features, pixels, inverse-dynamics (Pathak et al., 2017) and variational auto-encoders (Kingma & Welling, 2013) and evaluate generalization to unseen environments. (c) Analysis of some limitations of a direct prediction-error based curiosity formulation. We observe that if the agent itself is the source of stochasticity in the environment, it can reward itself without making any actual progress. We empirically demonstrate this limitation in a 3D navigation task where the agent controls different parts of the environment.

## 2 DYNAMICS-BASED CURIOSITY-DRIVEN LEARNING

Consider an agent that sees an observation $x_t$, takes an action $a_t$ and transitions to the next state with observation $x_{t+1}$. We want to incentivize this agent with a reward $r_t$ relating to how informative

the transition was. To provide this reward, we use an exploration bonus involving the following elements: (a) a network to embed observations into representations $\phi(x)$, (b) a forward dynamics network to predict the representation of the next state conditioned on the previous observation and action $p(\phi(x_{t+1})|x_t, a_t)$. Given a transition tuple $\{x_t, x_{t+1}, a_t\}$, the exploration reward is then defined as $r_t = -\log p(\phi(x_{t+1})|x_t, a_t)$, also called the *surprisal* (Achiam & Sastry, 2017).

An agent trained to maximize this reward will favor transitions with high prediction error, which will be higher in areas where the agent has spent less time, or in areas with complex dynamics. Such a dynamics-based curiosity has been shown to perform quite in some cases (Pathak et al., 2017), especially when the dynamics are learned in an embedding space rather than raw observations. In this paper, we explore dynamics-based curiosity further. We use mean-squared error corresponding to a fixed-variance Gaussian density as surprisal, i.e., $\|f(x_t, a_t) - \phi(x_{t+1})\|_2^2$ where f is the learned dynamics model. However, any other density model could be used.

## 2.1 FEATURE SPACES FOR FORWARD DYNAMICS

Consider the representation $\phi$ in the curiosity formulation above. If $\phi(x) = x$, the forward dynamics model makes predictions in the observation space. A good choice of feature space can make the prediction task more tractable and filter out irrelevant aspects of the observation space. But what makes a good feature space for dynamics driven curiosity? We propose the qualities that a good feature space must have:

- *Compactness*: The features should be easy to model by being low(er)-dimensional and filtering out irrelevant parts of the observation space.
- *Sufficiency*: The features should contain all the important information. Otherwise, the agent may fail to be rewarded for exploring some relevant aspect of the environment.
- *Stability*: Non-stationary rewards make it difficult for reinforcement agents to learn. Exploration bonuses by necessity introduce non-stationarity since what is new and novel becomes old and boring with time. In a dynamics-based curiosity formulation, there are two sources of non-stationarity: the forward dynamics model is evolving over time as it is trained and the features are changing as they learn. The former is intrinsic to the method, and the latter should be minimized where possible

In this work, we systematically investigate the efficacy of a number of feature-learning methods, summarized briefly as follows:

**Pixels** The simplest case is where $\phi(x) = x$ and we fit our forward dynamics model in the observation space. Pixels are sufficient, since no information has been thrown away, and stable since there is no feature learning component. However, learning from pixels is tricky because the observation space may be high-dimensional and complex.

**Random Features (RF)** The next simplest case is where we take our embedding network, a convolutional network, and fix it after random initialization. Because the network is fixed, the features are stable. The features can be made compact in dimensionality, but they are not constrained to be. However, random features may fail to be sufficient.

**Variational Autoencoders (VAE)** VAEs were introduced in (Kingma & Welling, 2013; Rezende et al., 2014) to fit latent variable generative models $p(x, z)$ for observed data $x$ and latent variable $z$ with prior $p(z)$ using variational inference. The method calls for an inference network $q(z|x)$ that approximates the posterior $p(z|x)$. This is a feedforward network that takes an observation as input and

|  | VAE | IDF | RF | Pixels |
|---|---|---|---|---|
| Stable | No | No | Yes | Yes |
| Compact | Yes | Yes | Maybe | No |
| Sufficient | Yes | Maybe | Maybe | Yes |

Table 1: Table summarizing the categorization of different kinds of feature spaces considered.

outputs a mean and variance vector describing a Gaussian distribution with diagonal covariance. We can then use the mapping to the mean as our embedding network $\phi$. These features will be a low-dimensional approximately sufficient summary of the observation, but they may still contain some irrelevant details such as noise, and the features will change over time as the VAE trains.

**Inverse Dynamics Features (IDF)**   Given a transition $(s_t, s_{t+1}, a_t)$ the inverse dynamics task is to predict the action $a_t$ given the previous and next states $s_t$ and $s_{t+1}$. Features are learned using a common neural network $\phi$ to first embed $s_t$ and $s_{t+1}$. The intuition is that the features learned should correspond to aspects of the environment that are under the agent's immediate control. This feature learning method is easy to implement and in principle should be invariant to certain kinds of noise (see (Pathak et al., 2017) for a discussion). A potential downside could be that the features learned may not be sufficient, that is they do not represent important aspects of the environment that the agent cannot immediately affect.

A summary of these characteristics is provided in Table 1. Note that the learned features are not stable because their distribution changes as learning progresses. One way to achieve stability could be to pre-train VAE or IDF networks. However, unless one has access to the internal state of the game, it is not possible to get a representative data of the game scenes to train the features. One way is to act randomly to collect data, but then it will be biased to where the agent started, and won't generalize further. Since all the features involve some trade-off of desirable properties, it becomes an empirical question as to how effective each of them is across environments.

## 2.2 Practical considerations in training an agent driven purely by curiosity

Deciding upon a feature space is only first part of the puzzle in implementing a practical system. Here, we detail the critical choices we made in the learning algorithm. Our goal was to reduce non-stationarity in order to make learning more stable and consistent across environments. Through the following considerations outlined below, we are able to get exploration to work reliably for different feature learning methods and environments with minimal changes to the hyper-parameters.

- *PPO*. In general, we have found the PPO algorithm (Schulman et al., 2017) to be a robust learning algorithm that requires little hyper-parameter tuning and so we stick to it for our experiments.

- *Reward normalization*. Since the reward function is non-stationary, it is useful to normalize the scale of the rewards so that the value function can learn quickly. We did this by dividing the rewards by a running estimate of the standard deviation of the sum of discounted rewards.

- *Advantage normalization*. While training with PPO, we normalize the advantages (Sutton & Barto, 1998) in a batch to have a mean of 0 and a standard deviation of 1.

- *Observation normalization*. We run a random agent on our target environment for 10000 steps, then calculate the mean and standard deviation of the observation and use these to normalize the observations when training. This is useful to ensure that the features do not have very small variance at initialization, and also ensure features have less variation across different environments.

- *More actors*. The stability of the method is greatly increased by increasing the number of parallel actors (which affects the batch-size) used. We typically use 128 parallel runs of the same environment for data collection while training an agent.

- *Normalizing the features*. In combining intrinsic and extrinsic rewards, we found it useful to ensure that the scale of the intrinsic reward was consistent across state space. We achieved this by using batch-normalization (Ioffe & Szegedy, 2015) in the feature embedding network.

## 2.3 'Death is not the end': discounted curiosity with infinite horizon

One important point is that the use of an end-of-episode signal, sometimes called 'done', can often leak information about the true reward function (assuming, as is common, that we have access to an extrinsic reward signal that we hide from the agent to measure pure exploration). If we don't remove the 'done' signal, many of the Atari games become too simple. For example, a simple strategy of giving +1 artificial reward at every time-step when the agent is alive and 0 upon death is sufficient to obtain a high score in some games, e.g. the Atari game 'Breakout' where it will seek to maximize the episode length and hence its score. In the case of negative rewards, the agent will try to end the episode as quickly as possible.

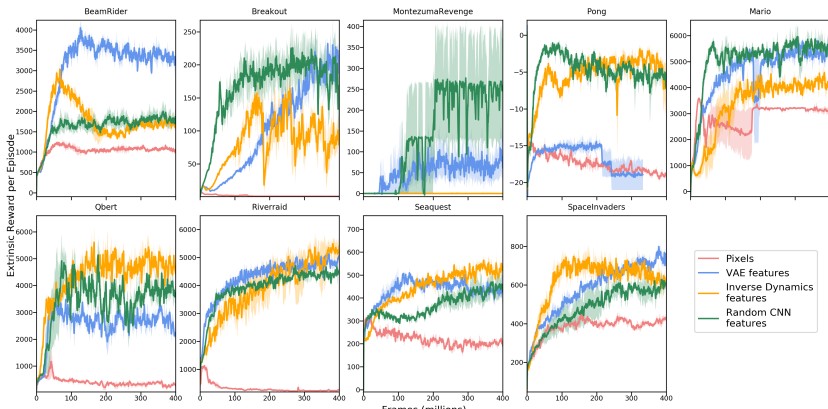

Figure 2: A comparison of feature learning methods on 8 selected Atari games and the Super Mario Bros. These evaluation curves show the mean reward (with standard error) of agents trained purely by curiosity, without reward or an end-of-episode signal. We see that our purely curiosity-driven agent is able to gather rewards in these environments without using any extrinsic reward at training. Results on all of the Atari games are in the appendix in Figure 8. We find curiosity model trained on pixels does not work well across any environment and VAE features perform either same or worse than random and inverse dynamics features. Further, inverse dynamics-trained features perform better than random features in 55% of the Atari games. An interesting outcome of this analysis is that random features for modeling curiosity are a simple, yet surprisingly strong baseline and likely to work well in half of the Atari games.

In light of this, if we want to study the behavior of a pure exploration agent, we should not bias it. In the infinite horizon setting (i.e., the discounted returns are not truncated at the end of the episode and always bootstrapped using the value function), death is just another transition to the agent, to be avoided only if it is "boring". Therefore, we removed 'done' to separate the gains of an agent's exploration from merely that of the death signal. In practice, we do find that the agent avoids dying in the games since that brings it back to the beginning of the game – an area it has already seen many times and where it can predict the dynamics well. This subtlety has been neglected by previous works showing experiments without extrinsic rewards.

## 3 EXPERIMENTS

In all of our experiments, both the policy and the embedding network work directly from pixels. For our implementation details including hyper-parameters and architectures, please refer to the Appendix A. Unless stated otherwise, all curves are the average of three runs with different seeds, and the shaded areas are standard errors of the mean. We have released the code and videos of a purely curious agent playing across all environments on our website.

### 3.1 CURIOSITY-DRIVEN LEARNING WITHOUT EXTRINSIC REWARDS

We begin by scaling up a pure curiosity-driven learning to a large number of environments without using any extrinsic rewards. We pick a total of 54 diverse simulated environments, as shown in Figure 1, including 48 Atari games, Super Mario Bros., 2 Roboschool scenarios (learning Ant controller and Juggling), Two-player Pong, 2 Unity mazes (with and without a TV controlled by the agent). The goal of this large-scale analysis is to investigate the following questions: (a) What happens when you run a pure curiosity-driven agent on a variety of games without any extrinsic rewards? (b) What kinds of behaviors can you expect from these agents? (c) What is the effect of the different feature-learning variants in dynamics-based curiosity on these behaviors?

**Atari Games**    To answer these questions, we began with a collection of well-known Atari games and ran a suite of experiments with different feature-learning methods. One way to measure how well a purely curious agent performs is to measure the extrinsic reward it is able to achieve, i.e. how good is the agent at playing the game. We show the evaluation curves of mean extrinsic reward in on

8 common Atari games in Figure 2 and all 48 Atari suite in Figure 8 in the appendix. It is important to note that the extrinsic reward is only used for evaluation, not for training. However, this is just a proxy for pure exploration because the game rewards could be arbitrary and might not align at all with how the agent explores out of curiosity.

The first thing to notice from the curves is: *most of them are going up*. This shows that a pure curiosity-driven agent can often learn to obtain external rewards without seeing any extrinsic rewards during training! To understand why this is happening, consider the game 'Breakout'. The main control action of the game is to keep hitting the bouncing ball with the paddle, but this does not earn any points. The game score increases only when the ball hits a brick (which then disappears). But the more bricks are struck by the ball, the more complicated the pattern of remaining bricks becomes, making the agent more curious to explore further, hence, collecting points as a bi-product. Furthermore, when the agent runs out of lives, the bricks are reset to the initial configuration, which has been seen by the agent many times before and is hence very predictable, so the agent tries to increase curiosity by staying alive and avoiding the death reset.

The fact that the curiosity reward is often sufficient is an unexpected result and might suggest that many popular RL test-beds do not need an external reward at all. It is likely that game designers (similar to architects, urban planners, gardeners, etc.) are purposefully setting up curricula to guide agents through the task by curiosity alone. This could explain why curiosity-like objective aligns reasonably well with the extrinsic reward in many human-designed environments (Lazzaro, 2004; Costikyan, 2013; Hunicke et al., 2004; Wouters et al., 2011). However, this is not always the case, and sometimes a curious agent can even do worse than a random agent. This happens when the extrinsic reward has little correlation with the agent's exploration, or when the agent fails to explore efficiently (e.g. see games 'Atlantis' and 'IceHockey' in Figure 8). We encourage the reader to refer to the game-play videos of the agent available on the website for a better understanding of the learned skills.

**Comparison of feature learning methods:** We compare four feature learning methods in Figure 2: raw pixels, random features, inverse dynamics features and VAE features. Training dynamics on raw pixels performs poorly across all the environments, while encoding pixels into features does better. This is likely because it is hard to learn a good dynamics model in pixel space, and prediction errors may be dominated by small irrelevant details.

Surprisingly, random features (RF) perform quite well across tasks and sometimes better than using learned features. One reason for good performance is that the random features are kept frozen (stable), the dynamics model learned on top of them has an easier time because of the stationarity of the target. In general, random features should work well in the domains where visual observations are simple enough, and random features can preserve enough information about the raw signal, for instance, Atari games. One scenario where IDF features consistently outperform random features is for generalization, e.g. training on one level of Mario Bros and testing on another (see Section 3.2 for details).

The VAE method also performed well but was somewhat unstable, so we decided to use RF and IDF for further experiments. The detailed result in appendix Figure 8 compares IDF vs. RF across the full Atari suite. To quantify the learned behaviors, we compared our curious agents to a randomly acting agent. We found that an IDF-curious agent collects more game reward than a random agent in 75% of the Atari games, an RF-curious agent does better in 70%. Further, IDF does better than RF in 55% of the games. Overall, random features and inverse dynamics features worked well in general. Further details in the appendix.

**Super Mario Bros.** We compare different feature-learning methods in Mario Bros in Figure 2. Super Mario Bros has already been studied in the context of extrinsic reward free learning (Pathak et al., 2017) in small-scale experiments, and so we were keen to see how far curiosity alone can push the agent. We used a more efficient version of the Mario simulator, allowing for longer training, while keeping observation space, actions, and dynamics of the game the same. Due to 100x longer training and using PPO for optimization, our agent was able to pass several levels of the game, significantly improving over prior exploration results on Mario Bros.

Could we further push the performance of a purely curious agent by making the underlying optimization more stable? One way is to scale up the batch-size. We do so by increasing the number of parallel threads for running environments from 128 to 1024. We show the comparison between

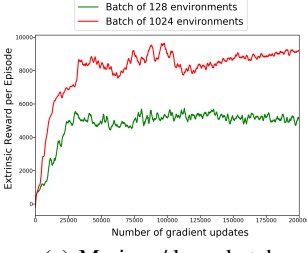 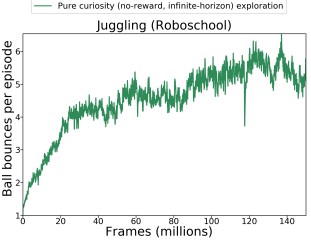 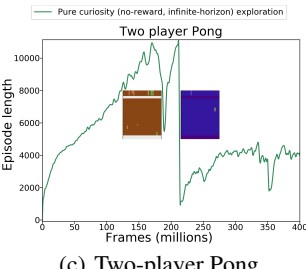

(a) Mario w/ large batch     (b) Juggling (Roboschool)     (c) Two-player Pong

Figure 3: (a) Left: A comparison of the RF method on Mario with different batch sizes. Results are without using extrinsic reward. (b) Center: Number of ball bounces in the Juggling (Roboschool) environment. (c) Right: Mean episode length in the multiplayer Pong environment. The discontinuous jump on the graph corresponds to the agent reaching a limit of the environment - after a certain number of steps in the environment the Atari Pong emulator starts randomly cycling through background colors and becomes unresponsive to agent's actions

training using 128 and 1024 parallel environment threads in Figure 3(a). As apparent from the graph, training with large batch-size using 1024 parallel environment threads performs much better. In fact, the agent is able to explore much more of the game: *discovering 11 different levels of the game*, finding secret rooms and defeating bosses. Note that the x-axis in the figure is the number of gradient steps, not the number of frames, since the point of this large-scale experiment is not a claim about sample-efficiency, but performance with respect to training the agent. This result suggests that the performance of a purely curiosity-driven agent would improve as the training of base RL algorithm (PPO in our case) gets better. The video is on the website.

**Roboschool Juggling**  We modified the Pong environment from the Roboschool framework to only have one paddle and to have two balls. The action space is continuous with two-dimensions, and we discretized the action space into 5 bins per dimension giving a total of 25 actions. Both the policy and embedding network are trained on pixel observation space (note: not state space). This environment is more difficult to control than the toy physics used in games, but the agent learns to intercept and strike the balls when it comes into its area. We monitored the number of bounces of the balls as a proxy for interaction with the environment, as shown in Figure 3(b). See the video on the project website.

**Roboschool Ant Robot**  We also explored using the Ant environment which consists of an Ant with 8 controllable joints on a track. We again discretized the action space and trained policy and embedding network on raw pixels (not state space). However, in this case, it was less easy to measure exploration because the extrinsic distance reward measures progress along the racetrack, but a purely curious agent is free to move in any direction. We find that a walking like behavior emerges purely out of a curiosity-driven training. We refer the reader to the result video showing that the agent is meaningfully interacting with the environment.

**Multi-agent curiosity in Two-player Pong**  We have already seen that a purely curiosity-driven agent learns to play several Atari games without reward, but we wonder how much of that behavior is caused by the fact that the opposing player is a computer agent with a hard-coded strategy. What would happen if we were to make both the players curious-driven? To find out, we set up a two-player Pong game where both the sides (paddles) of the game are controlled by two curiosity-driven agents. We shared the initial layers of both the agents but have different action heads, i.e., total action space is now the cross product of the actions of player 1 by the actions of player 2.

Note that the extrinsic reward is meaningless in this context since the agent is playing both sides, so instead, we show the length of the episode. The results are shown in Figure 3(c). We see from the episode length that the agent learns to have longer rallies over time, learning to play pong without any teacher – purely by curiosity on both sides. In fact, *the game rallies eventually get so long that they break our Atari emulator* causing the colors to change radically, which crashes the policy as shown in the plot.

## 3.2 GENERALIZATION ACROSS NOVEL LEVELS IN SUPER MARIO BROS.

In the previous section, we showed that our purely curious agent can learn to explore efficiently and learn useful skills, e.g., game playing behaviour in games, walking behaviour in Ant etc. So far, these skills were shown in the environment where the agent was trained on. However, one advantage of developing reward-free learning is that one should then be able to utilize abundant "unlabeled" environments without reward functions by showing generalization to novel environments.

To test this, we first pre-train our agent using curiosity only in the Level 1-1 of Mario Bros. We investigate how well RF and IDF-based curiosity agents generalize to novel levels of Mario. In Figure 4, we show two examples of training on one level of Mario and fine-tuning on another testing level, and compare to learning on the testing level from scratch. The training signal in all the cases is curiosity-only reward. In the first case, from Level 1-1 to Level 1-2, the global statistics of the environments match (both are 'day-time' environments, i.e., blue sky) but levels have different enemies, different geometry, and different difficulty. We see that there is strong transfer from for both methods in this scenario. However, the transfer performance is weaker in the second scenario from Level 1-1 to Level 1-3. This is so because the problem is considerably harder for the latter level pairing as there is a color pallette shift from day to night, as shown in Figure 4.

We further note that IDF-learned features transfer in both the cases and random features transfer in the first case, but do not transfer in the second scenario from day to night. These results might suggest that while random features perform well on training environments, learned features appear to generalize better to novel levels. However, this needs more analysis in the future across a large variety of environments. Overall, we find some promising evidence showing that skills learned by curiosity help our agent explore efficiently in novel environments.

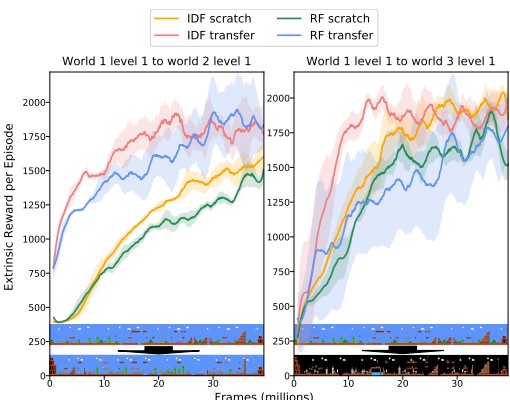

Figure 4: Mario generalization experiments. On the left we show transfer results from Level 1-1 to Level 1-2, and on the right we show transfer results from Level 1-1 to Level 1-3. Underneath each plot is a map of the source and target environments. All agents are trained without extrinsic reward.

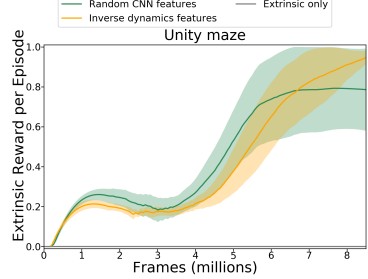

Figure 5: Mean extrinsic reward in the Unity environment while training with terminal extrinsic + curiosity reward. Note that the curve for extrinsic reward only training is constantly zero.

## 3.3 CURIOSITY WITH SPARSE EXTERNAL REWARD

In all our experiments so far, we have shown that our agents can learn useful skills without any extrinsic rewards, driven purely by curiosity. However, in many scenarios, we might want the agent to perform some particular task of interest. This is usually conveyed to the agent by defining extrinsic rewards. When rewards are dense (e.g. game score at every frame), classic RL works well and intrinsic rewards generally should not help performance. However, designing dense rewards is a challenging engineering problem (see introduction for details). In this section, we evaluate how well curiosity can help an agent perform a task in presence of sparse, or just terminal, rewards.

**Terminal reward setting**: For many real problems, only terminal reward is available, e.g. in navigation, you only get rewards once you find what you were looking for. This is a setting where classic RL typically performs poorly. Hence, we consider the 3D navigation in a maze designed

in the Unity ML-agent framework with 9 rooms and a sparse terminal reward. The action space is discrete, consisting of: move forward, look left 15 degrees, look right 15 degrees and no-op. The agent starts in room-1, which is furthest away from room-9 which contains the goal. We compare an agent trained with extrinsic reward (+1 when the goal is reached, 0 otherwise) to an agent trained with extrinsic + intrinsic reward. Extrinsic only (classic RL) never finds the goal in all our trials, which means it is impossible to get any meaningful gradients. Whereas extrinsic+intrinsic typically converges to getting the reward every time. Results in Figure 5 show results for vanilla PPO, PPO + IDF-curiosity and PPO + RF-curiosity.

**Sparse reward setting**: In preliminary experiments, we picked 5 Atari games which have sparse rewards (as categorized by (Bellemare et al., 2016)), and compared extrinsic (classic RL) vs. extrinsic+intrinsic (ours) reward performance. In 4 games out of 5, curiosity bonus improves performance (see Table 2 in the appendix, the higher score is better). We would like to emphasize that this is not the focus of the paper, and these experiments are provided just for completeness. We just combined extrinsic (coefficient 1.0) and intrinsic reward (coefficient 0.01) directly without any tuning. We leave the question on how to optimally combine extrinsic and intrinsic rewards as a future direction.

## 4   RELATED WORK

**Intrinsic Motivation:** A family of approaches to intrinsic motivation reward an agent based on prediction error (Schmidhuber, 1991c; Stadie et al., 2015; Pathak et al., 2017; Achiam & Sastry, 2017), prediction uncertainty (Still & Precup, 2012; Houthooft et al., 2016), or improvement (Schmidhuber, 1991a; Lopes et al., 2012) of a forward dynamics model of the environment that gets trained along with the agent's policy. As a result the agent is driven to reach regions of the environment that are difficult to predict for the forward dynamics model, while the model improves its predictions in these regions. This adversarial and non-stationary dynamics can give rise to complex behaviors. Relatively little work has been done in this area on the pure exploration setting where there is no external reward. Of these mostly closely related are those that use a forward dynamics model of a feature space such as Stadie et al. (2015) where they use autoencoder features, and Pathak et al. (2017) where they use features trained with an inverse dynamics task. These correspond roughly to the VAE and IDF methods detailed in Section 2.1.

Smoothed versions of state visitation counts can be used for intrinsic rewards (Bellemare et al., 2016; Fu et al., 2017; Ostrovski et al., 2017; Tang et al., 2017). Count-based methods have already shown very strong results when combining with extrinsic rewards such as setting the state of the art in the Atari game Montezuma's Revenge (Bellemare et al., 2016), and also showing significant exploration of the game without using the extrinsic reward. It is not yet clear in which situations count-based approaches should be preferred over dynamics-based approaches; we chose to focus on dynamics-based bonuses in this paper since we found them straightforward to scale and parallelize. In our preliminary experiments, we did not have sufficient success with already existing count-based implementations in scaling up for a large-scale study.

Learning without extrinsic rewards or fitness functions has also been studied extensively in the evolutionary computing where it is referred to as 'novelty search' (Lehman & Stanley, 2008; 2011; Stanley & Lehman, 2015). There the novelty of an event is often defined as the distance of the event to the nearest neighbor amongst previous events, using some statistics of the event to compute distances. One interesting finding from this literature is that often much more interesting solutions can be found by not solely optimizing for fitness.

Other methods of exploration are designed to work in combination with maximizing a reward function, such as those utilizing uncertainty about value function estimates (Osband et al., 2016; Chen et al., 2017), or those using perturbations of the policy for exploration (Fortunato et al., 2017; Plappert et al., 2017). Schmidhuber (2010) and Oudeyer & Kaplan (2009); Oudeyer (2018) provide a great review of some of the earlier work on approaches to intrinsic motivation. Alternative methods of exploration include Sukhbaatar et al. (2018) where they utilize an adversarial game between two agents for exploration. In Gregor et al. (2017), they optimize a quantity called empowerment which is a measurement of the control an agent has over the state. In a concurrent work, diversity is used as a measure to learn skills without reward functions Eysenbach et al. (2018).

**Random Features:** One of the findings in this paper is the surprising effectiveness of random features, and there is a substantial literature on random projections and more generally randomly

initialized neural networks. Much of the literature has focused on using random features for classification (Saxe et al., 2011; Jarrett et al., 2009; Yang et al., 2015) where the typical finding is that whilst random features can work well for simpler problems, feature learning performs much better once the problem becomes sufficiently complex. Whilst we expect this pattern to also hold true for dynamics-based exploration, we have some preliminary evidence showing that learned features appear to generalize better to novel levels in Mario Bros.

## 5 DISCUSSION

We have shown that our agents trained purely with a curiosity reward are able to learn useful behaviours: (a) Agent being able to play many Atari games without using any rewards. (b) Mario being able to cross over 11 levels without any extrinsic reward. (c) Walking-like behavior emerged in the Ant environment. (d) Juggling-like behavior in Robo-school environment (e) Rally-making behavior in Two-player Pong with curiosity-driven agent on both sides. But this is not always true as there are some Atari games where exploring the environment does not correspond to extrinsic reward.

More generally, our results suggest that, in many game environments designed by humans, the extrinsic reward is often aligned with the objective of seeking novelty.

**Limitation of prediction error based curiosity:** A more serious potential limitation is the handling of stochastic dynamics. If the transitions in the environment are random, then even with a perfect dynamics model, the expected reward will be the entropy of the transition, and the agent will seek out transitions with the highest entropy. Even if the environment is not truly random, unpredictability caused by a poor learning algorithm, an impoverished model class or partial observability can lead to exactly the same problem. We did not observe this effect in our experiments on games so we designed an environment to illustrate the point.

We return to the maze of Section 3.3 to empirically validate a common thought experiment called the noisy-TV problem. The idea is that local sources of entropy in an environment like a TV that randomly changes channels when an action is taken should prove to be an irresistible attraction to our agent. We take this thought experiment literally and add a TV to the maze along with an action to change the channel. In Figure 6 we show how adding the noisy-TV affects the performance of IDF and RF. As expected the presence of the TV drastically slows down learning, but we note that if you run the experiment for long enough the agents do sometimes converge to getting the extrinsic reward consistently. We have shown empirically that stochasticity can be a problem, and so it is important for future work to address this issue in an efficient manner.

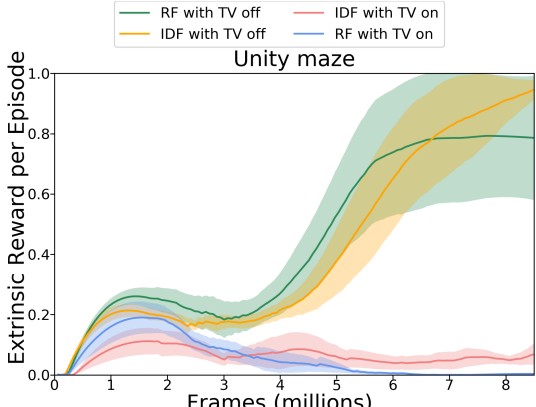

Figure 6: We add a noisy TV to the unity environment in Section 3.3. We compare IDF and RF with and without the TV.

**Future Work:** We have presented a simple and scalable approach that can learn nontrivial behaviors across a diverse range of environments without any reward function or end-of-episode signal. One surprising finding of this paper is that random features perform quite well, but learned features appear to generalize better. While we believe that learning features will become more important once the environment is complex enough, we leave that for future work to explore.

Our wider goal, however, is to show that we can take advantage of many unlabeled (i.e., not having an engineered reward function) environments to improve performance on a task of interest. Given this goal, showing performance in environments with a generic reward function is just the first step, and future work will hopefully investigate transfer from unlabeled to labeled environments.

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

## A    IMPLEMENTATION DETAILS

We have released the training code and environments on our website [1]. For full details, we refer the reader to our code and video results in the website.

**Pre-processing:**    All experiments were done with pixels. We converted all images to grayscale and re-sized to size 84x84. We learn the agent's policy and forward dynamics function both on a stack of historical observations $[x_{t-3}, x_{t-2}, x_{t-1}, x_t]$ instead of only using the current observation. This is to capture partial observability in these games. In the case of Super Mario Bros and Atari experiments, we also used a standard frameskip wrapper that repeats each action 4 times.

**Architectures:**    Our embedding network and policy networks had identical architectures and were based on the standard convolutional networks used in Atari experiments. The layer we take as features in the embedding network had dimension 512 in all experiments and no nonlinearity. To keep the scale of the prediction error consistent relative to extrinsic reward, in the Unity experiments we applied batchnorm to the embedding network. We also did this for the Mario generalization experiments to reduce covariate shift from level to level. For the VAE auxiliary task and pixel method, we used a similar deconvolutional architecture the exact details of which can be found in our code submission. The IDF and forward dynamics networks were heads on top of the embedding network with several extra fully-connected layers of dimensionality 512.

**Hyper-parameters:**    We used a learning rate of 0.0001 for all networks. In most experiments, we used 128 parallel environments with the exceptions of the Unity and Roboschool experiments where we could only run 32 parallel environments, and the large scale Mario experiment where we used 1024. We used rollouts of length 128 in all experiments except for the Unity experiments where we used 512 length rollouts so that the network could quickly latch onto the sparse reward. In the initial 9 experiments on Mario and Atari, we used 3 optimization epochs per rollout in the interest of speed. In the Mario scaling, generalization experiments, as well as the Roboschool experiments, we used 6 epochs. In the Unity experiments, we used 8 epochs, again to more quickly take advantage of sparse rewards.

## B    ADDITIONAL RESULTS

### B.1    ATARI

To better measure the amount of exploration, we provide the best return of curiosity-driven agents in figure 7(a) and the episode lengths in figure 7(b). Notably on Pong the increasing episode length combined with a plateau in returns shows that the agent maximizes the number of ball bounces, rather than the reward.

Figure 8 shows the performance of curiosity-driven agents based on Inverse Dynamics and Random features on 48 Atari games.

Although not the focus of this paper, for completeness we include some results on combining intrinsic and extrinsic reward on several sparse reward Atari games. When combining with extrinsic rewards, we use the end of the episode signal. The reward used is the extrinsic reward plus 0.01 times the intrinsic reward. The results are shown in Table 2. We don't observe a large difference between the settings, likely because the combination of intrinsic and extrinsic reward needs to be tuned. We did observe that one of the intrinsic+extrinsic runs on Montezuma's Revenge explored 10 rooms.

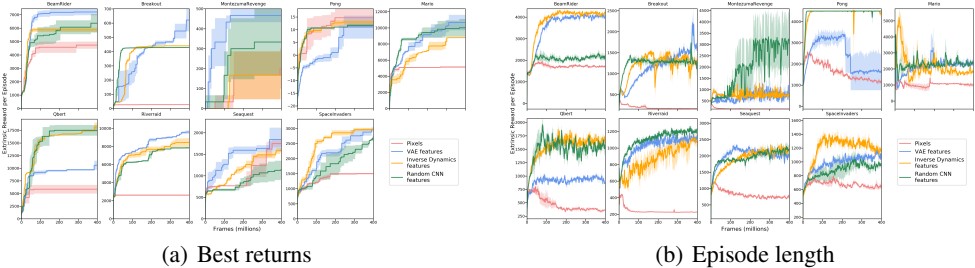

(a) Best returns                                    (b) Episode length

Figure 7: (a) Left: Best extrinsic returns on eight Atari games and Mario. (c) Right: Mean episode lengths on eight Atari games and Mario.

---

[1]Website at https://doubleblindsupplementary.github.io/large-curiosity/

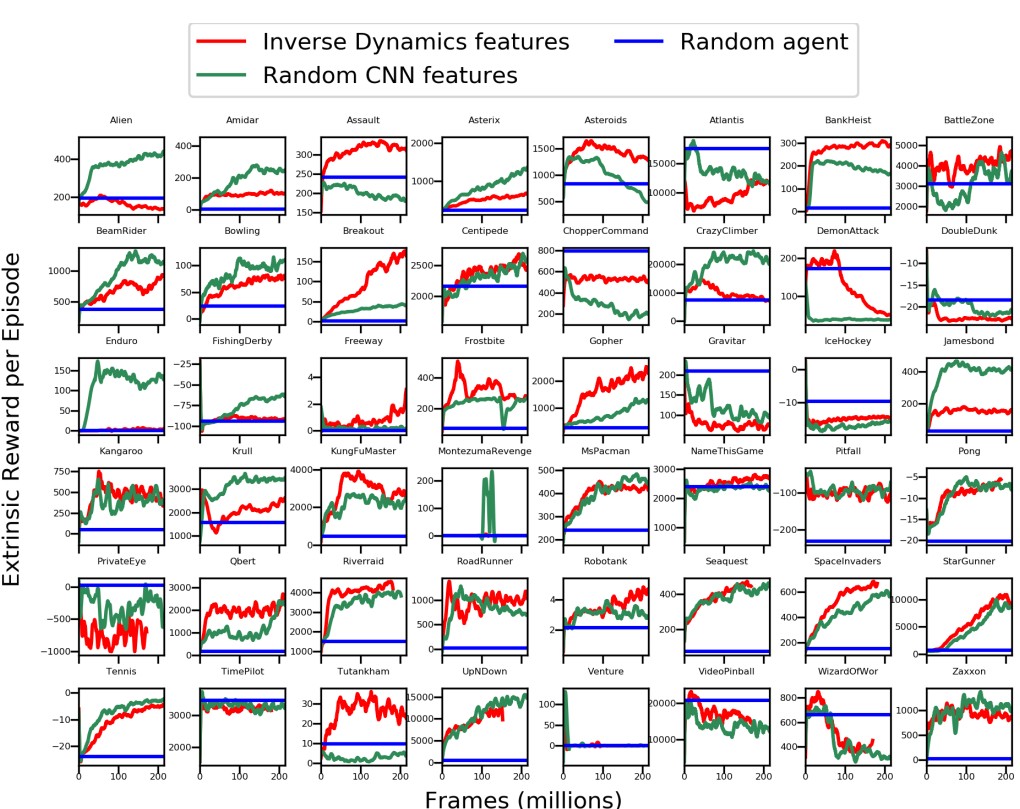

Figure 8: Pure curiosity-driven exploration (no extrinsic reward, or end-of-episode signal) on 48 Atari games. We observe that the extrinsic returns of curiosity-driven agents often increases despite the agents having no access to the extrinsic return or end of episode signal. In multiple environments, the performance of the curiosity-driven agents is significantly better than that of a random agent, although there are environments where the behavior of the agent is close to random, or in fact seems to minimize the return, rather than maximize it. For the majority of the training process RF perform better than a random agent in about 67% of the environments, while IDF perform better than a random agent in about 71% of the environments.

| Reward | Gravitar | Freeway | Venture | PrivateEye | MontezumaRevenge |
|---|---|---|---|---|---|
| Ext Only | $999.3 \pm 220.7$ | $33.3 \pm 0.6$ | $0 \pm 0$ | $5020.3 \pm 395$ | $1783 \pm 691.7$ |
| Ext + Int | $1165.1 \pm 53.6$ | $32.8 \pm 0.3$ | $416 \pm 416$ | $3036.5 \pm 952.1$ | $2504.6 \pm 4.6$ |

Table 2: These results compare the mean reward ($\pm$ std-error) after 100 million frames across 3 seeds for an agent trained with intrinsic plus extrinsic reward versus extrinsic reward only. The extrinsic (coefficient 1.0) and intrinsic reward (coefficient 0.01) were directly combined without any hyper-parameter tuning. We leave the question on how to optimally combine extrinsic and intrinsic rewards up to future work. This is to emphasize that combining extrinsic with intrinsic rewards is not the focus of the paper, and these experiments are provided just for completeness.

## B.2 MARIO

We show the analogue of the plot shown in Figure 3(a) showing max extrinsic returns. See Figure 9.

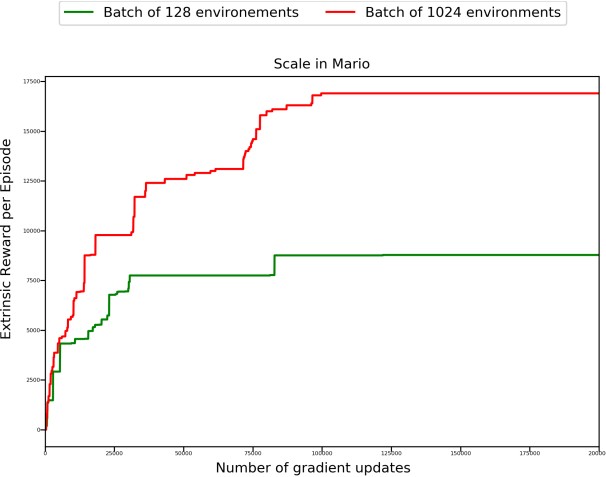

Figure 9: Best extrinsic returns on the Mario scaling experiments. We observe that larger batches allow the agent to explore more effectively, reaching the same performance in less parameter updates, and also achieving better ultimate scores.

---

**Algorithm 1:** Curiosity-driven Learning

---

1   Initialize the networks $f(x_t, a_t; \theta_f)$, $\pi(x_t; \theta_\pi)$ and $\phi(x; \theta_\phi)$

2   $D = \{\}$

3   **for** *iteration i = 1 to . . .* **do**

4      **for** *envs in parallel t = 1 to 128* **do**

5        **for** *iteration t = 1 to 128* **do**

6          Sample $a \sim \pi(x_t; \theta_\pi)$ and act using $a$ in the environment

7          $D \Leftarrow D + (x_t, a_t, x_{t+1}, r_t)$ where $r_t = \|f(x_t, a_t; \theta_f) - \phi(x_{t+1}; \theta_\phi)\|_2^2$

8        **end**

9      **end**

10     **for** *steps k = 1 to 64* **do**

11       Sample batch size of 2048 from $D$ and update using ADAM as follows:

12         $\theta'_f := \theta_f - \eta_1 \, \nabla_{\theta_f} \mathbb{E}\big[\|f(x_t, a_t; \theta_f) - \phi(x_{t+1}; \theta_\phi)\|_2^2\big]$

13         $\theta'_\phi := \theta_\phi - \eta_2 \, \nabla_{\theta_\phi} \mathbb{E}\big[\| \dots \|_2^2\big]$: some auxiliary task

14         $\theta'_\pi := \theta_\pi + \eta_3 \, \nabla_{\theta_\pi} \mathbb{E}_{\pi(x_t; \theta_\pi)}\big[\sum_t r_t\big]$: use PPO with discounted returns

15       $\theta_f \Leftarrow \theta'_f$

16       $\theta_\phi \Leftarrow \theta'_\phi$

17       $\theta_\pi \Leftarrow \theta'_\pi$

18     **end**

19   **end**

---

