# OpenReview forum: "Large-Scale Study of Curiosity-Driven Learning"
_ICLR.cc/2019/Conference_

### Official Review · AnonReviewer3 · 2018-11-02
**Nice experimental paper on curiosity based RL**

**Rating:** 7
**Confidence:** 3

**Review:**

In this paper, the authors presented a large experimental study of curiosity-driven reinforcement learning on various tasks. In the experimental studies, the authors also compared several feature space embedding methods, including identical mapping (pixels), random embedding, variational autoencoders and inverse dynamics features. The authors found that in many of the tasks, learning based on intrinsic rewards could generate good performance on extrinsic rewards, when the intrinsic rewards and extrinsic rewards are correlated. The authors also found that random features embedding, somewhat surprisingly, performs well in the tasks.

Overall, the paper is well written with clarity. Experimental setup is easy to understand. The authors provided code, which could help other researchers reproduce their result.

Weaknesses:

1) as an experimental study, it would be valuable to compare the performance of curiosity-based learning versus learning based on well-defined extrinsic rewards. The author is correct that in many tasks, well-behaved extrinsic rewards are hard to find. But for problems with well-defined extrinsic rewards, such a comparison could help readers understand the relative performance of curiosity-based learning and/or how much headroom there exists to improve the current methods.

2) it is surprising that random features perform so well in the experiments. The authors did provide literature in classification that had similar findings, but it would be beneficial for the authors to explore reasons that random features perform well in reinforcement learning.

---

> ### Author Response · Authors · 2018-11-22
> **[Authors' Response to R3] Comparison to extrinsic reward; discussing random features**
>
> We thank you for the constructive feedback and discuss some of your comments below.
>
> R3: "it would be valuable to compare the performance of curiosity-based learning versus learning based on well-defined extrinsic rewards"
> => We would like to highlight that evaluating success of pure curiosity-driven exploration (no extrinsic rewards for training) by measuring the extrinsic score of game is just a proxy to evaluate exploration. Our results show that exploration via curiosity has striking correlation with game scores. But we expect that when environments have a well-defined (and well-shaped!) extrinsic reward, a policy trained using that extrinsic reward should outperform the policy trained with only curiosity especially when the performance is measured by the extrinsic return.
>
> There are, however, examples, such as the Bowling Atari game, where a policy trained with only curiosity does *better* than a policy trained with extrinsic rewards. The purely curious agent learns to play the game better than agents trained to maximize the (clipped) extrinsic reward directly. We think this is because the agent gets attracted to the difficult-to-predict flashing of the scoreboard occurring after the strikes. We expect such examples to come from environments with misleading or poorly-shaped extrinsic rewards.
>
>
> R3: "...it would be beneficial for the authors to explore reasons that random features perform well in reinforcement learning."
> => In the paper, Section 2.1, we discuss that random features have advantages that they are they are stable, compact, and tend to include most relevant information about the observation. However, in our opinion, a more interesting question is not why random features perform so well, but rather why the feature learning methods perform so poorly (relative to this baseline). Learning the features introduces non-stationarity that confounds the effects of learning the dynamics. We believe that if methods are developed to address this non-stationarity, or environments that are more visually complex are used, then the benefits of the learning the features will become more noticeable.

---

### Official Review · AnonReviewer1 · 2018-11-02
**This paper conducts a large series of experiments on curiosity based rewards for RL agents, discuss different setups for the intrinsic reward, and, experiment on a wide range of tasks.**

**Rating:** 9
**Confidence:** 5

**Review:**

This paper studies the dynamics-based curiosity intrinsic reward where the agent is rewarded highly in states where the forward dynamic prediction errors are high in an embedding space (either due to complexity of the state or unfamiliarity).

Overall I like the paper, it's systematic and follows a series of practical considerations and step-by-step experimentations.

One of the main area which is missing in the paper is the comparison to two other class of RL methods: count-based exploration and novelty search. While the section 4 has a discussion on related papers, there's no systematic experimental comparison across these methods. In sec. 4, there's a reference to an initial set of experiments with count-based methods without much details.

Another area of improvement is the experiments around VAE. While the paper shows experimentally that they aren't as successful as the RFs or IDFs, there's no further discussion on the reasons for poor performance.

Also it's not clear from the details in the paper what are the architectures for the VAE and RFs (there's a reference to the code but would've been better to have sufficient details in the paper).

An interesting area for future work could be on early stopping techniques for embedding training - it seems that RFs perform well without any training while in some scenarios the IDFs work overall the best. So it would be interesting to explore how much training is needed for the embedding model. RFs are never trained and IDFs are continuously trained. So maybe somewhere in between could be the sweet spot with training for a short while and then fixing the features.

---

> ### Author Response · Authors · 2018-11-22
> **[Authors' Response to R1] Thank you**
>
> We thank you for the constructive feedback and are glad that you enjoyed the paper. Here we discuss some of your comments.
>
> R1: "missing in the paper is the comparison to two other class of RL methods: count-based exploration... In sec. 4, there's a reference to an initial set of experiments with count-based methods without much details."
> => We chose to focus on dynamics-based approaches in this paper because we found them more straightforward to efficiently parallelize than the published pseudo-count methods. This allows us to be able to run more and larger experiments on many environments. Interestingly, increased parallelization also significantly helped the exploration strategies as shown in Figure 3(a).
>
> Further, we will add the details of preliminary experiments using pseudo-count in the supplementary. In particular, we were not able to find any official public implementation of the pseudo-count methods. We experimented with a third party implementation trying to see if it could play Breakout without extrinsic rewards, but did not achieve sufficient success and found it to be too slow for scaling it up to a large-scale study.
>
>
> R1: "the experiments around VAE... While the paper shows experimentally that they aren't as successful... there's no further discussion on the reasons for poor performance."
> => We found that VAEs overall worked well and were sometimes better than other representation learning methods, but often were causing instability at training. We don't claim such instability is an inherent property of the VAE feature learning method, but probably stems from the continually changing data distribution as agent makes progress. Indeed modeling the density of a non-stationary distribution, with modes appearing and disappearing, is a challenging and an active research problem. We will clarify this in the final version.
>
> R1: "An interesting area for future work could be on early stopping techniques for embedding training… maybe somewhere in between could be the sweet spot with training"
> => Thank you for the excellent suggestion. We agree that there may be some optimal tradeoff between features that are stable and features that adapt to the environment. Such tradeoffs would be interesting to investigate, and might be crucial to getting learned features to perform significantly better than fixed random features. We will add this in the discussion/future work section of paper.
>
> R1: "What are the architectures for the VAE and RFs (there's a reference to the code but would've been better to have sufficient details in the paper)."
> => Thank you. We will add more details on the architectures to the appendix.

---

### Official Review · AnonReviewer2 · 2018-11-05
**Using curiosity-based reward exclusively works for game environments; not clear that this would be the case for more practical settings and findings regarding varying effectiveness of observation representation are largely incomplete. However, the core finding should influence additional research in game environments.**

**Rating:** 6
**Confidence:** 4

**Review:**

The authors consider the setting of a RL agent that exclusively receives intrinsic reward during training that is intended to model curiosity; technically, ‘curiosity’ is quantified by the ability of the agent to predict its own forward dynamics [Pathak, et al., ICML17]. This study primarily centers around an initially somewhat surprising result that non-trivial policies can be learned for many ’simpler’ video games (e.g., Atari, Super Mario, Pong) using just curiosity as reward. While this is primarily an empirical study, one aspect considered was the observation representation (raw pixels, random features, VAE, and inverse dynamics features [Pathak, et al., ICML17]). In examining reward curves (generally extrinsic during testing), ‘curiosity-based’ reward generally works with the representation effectiveness varying across different testbeds. They also conduct more in-depth experiments on specific testbeds to study the dynamics (e.g., Super Mario, Juggling, Ant Robot, Multi-agent Pong) — perhaps most interestingly showing representation-based transfer of different embeddings across levels in Super Mario. Finally, they consider the Unity maze testbed, combining intrinsic rewards with the end-state goal reward to generate a more dense reward space.

From a high level perspective, this is an interesting result that ostensibly will lead to a fair amount of discussion within the RL community (and already has based on earlier versions of this work). However, it isn’t entirely clear if the primary contribution is showing that ‘curiosity reward’ is a potentially promising approach or if game environments aren’t particularly good testbeds for practical RL algorithms — given the lack of significant results on more realistic domains, my intuition leans toward the later (the ant robot is interesting, but one can come up with ‘simulator artifact’ based explanations). And honestly, I think the paper reads as if leaning toward the same conclusion. Regardless, given the prevalence of these types of testbed environments, either is a useful discussion to have. Maybe the end result could minimally be a new baseline that can help quantify the ‘difficulty’ of a particular environment.

From the perspective of a purely technical contribution, there are fewer exciting results. The basic method is taken from [Parthak, et al., ICML17] (modulo some empirical choices such as using PPO). The comparison of different observation representations doesn’t include any analytical component, the empirical component is primarily inconclusive, and the position statements are fairly non-controversial (and not really conclusively supported). The testbeds all existed previously and this is mostly the effort of pulling then together. Even the ‘focused experiments’ can be explained with the intuitive narrative that in the state/action space, there is always more uncertainty the farther one goes from the starting point and this is more of a result of massive computation being applied primarily to problems that are designed to provide some level of novelly (the Roboschool examples are a bit more interesting, but also less conclusive). Finally, Figure 5 is interesting in showing that ‘curiosity + extrinsic’ improves over extrinsic rewards — although this isn’t particularly surprising for maze navigation that has such sparse rewards and can be viewed as something like ‘active exploration’. With respect to this specific setting, the authors may want to consider [Mirowski, et al., Learning to Navigate in Complex Environments, ICLR17] with respect to auxiliary loss + RL extrinsic rewards to improve performance (in this case, also in maze environments).

In just considering the empirical results, they clearly entail a fair amount of effort and just a dump of the code and experiments on the community will likely lead to new findings (even if they are that game simulators are weaker testbeds than previously thought). It is easy to ask for additional experiments (i.e., other mechanisms of uncertainty such as the count-based discussed in related work, other settings in 2.2) — but the quality seems high enough that I basically trust the settings and findings. Beyond the core findings, the other settings are less convincingly supported by seem more like work in progress and this paper is really just a scaling-up of [Pathak, et al., ICML17] without generating any strong results regarding questions around representation, what to do about stochasticity (although the discussion regarding something like ‘curiosity honeypots’ is interesting). Thus, it reads like one interesting finding around curiosity-driven RL working in games plus a bunch of preliminary findings trying to grasp at some explanations and potential future directions.

Evaluating the paper along the requested dimensions:

= Quality: The paper is well-written with a large set of experiments, making the case that exclusively using curiosity-based reward is very promising for the widely-used game RL testbeds. Modulo a few pointers, the work is well-contextualized and makes reasonable assumptions in conducting its experiments. The submitted code and videos result in a high-quality presentation and trustworthiness of the results. (7/10)

= Clarity: The paper is very clearly written. (7/10)

= Originality: The algorithmic approach is a combination of [Parthak, et al., ICML17] and [Schulman, et al. 2017] (with some experiments using [Kingma & Welling, 2013]). All of the testbeds have been used previously. Other than completely relying on curiously-based reward exclusively, there is little here. In considering combining with extrinsic rewards, I would also consider [Mirowski, et al., ICLR17], which is actually more involved in this regard. (4/10)

= Significance: Primarily, this ‘finishes’ [Parthak, et al., ICML17] to its logical conclusion for game-based environments and should spur interesting conversations and further research. In terms of actual technical contributions, I believe much less significant. (5/10)

=== Pros ===
+ demonstrates that curiosity-based reward works in simpler game environments
+ (implicitly) calls into question the value of these testbed environments
+ well written, with a large set of experiments and some interesting observations/discussions

=== Cons ===
- little methodological innovation or analytical explanations
- offers minimal (but some) evidence that curiosity-based reward works in more realistic settings
- doesn’t answer the one question regarding observation representation that it set out to evaluate
- the more interesting problem, RL + auxiliary loss isn’t evaluated in detail
- presumably, the sample complexity is ridiculous

Overall, I am ambivalent. I think that more casual ML/RL researchers will find these results controversial and surprising while more experienced researchers will see curiosity-driven learning to be explainable primarily by the intuition of the “The fact that the curiosity reward is often sufficient” paragraph of page 6, demanding more complex environments before accepting that this form of curiosity is particularly useful. The ostensible goal of learning more about observation representations is mostly preliminary — and this direction holds promise of for a stronger set of findings. Dealing with highly stochastic environments seems a potential fatal flaw of the assumptions of this method. However, as I said previously, this is probably a discussion worth having given the popularity and visibility of game-based testbeds — so, coupled with the overall quality of the paper, I lean toward a weak accept.

---

> ### Author Response · Authors · 2018-11-22
> **[Authors' Response to R2] Thank you; The usefulness of random features is demonstrated**
>
> We thank you for the  detailed and thoughtful review. We are glad that you found the paper well-contextualized and the presentation high-quality. Here we discuss some of your comments.
>
> R2: "this 'finishes' [Pathak et al., ICML17] to its logical conclusion for game-based environments and should spur interesting conversations and further research. In terms of actual technical contributions, I believe much less significant."
> => In the light of the comments on originality and significance, we would like to highlight our finding that random features perform quite well and at times as well as learned features across many environments. This is a novel contribution since prior works have relied on learned features as a crucial requirement for good performance [Pathak et. al. ICML17]. We believe this investigation would allow random features to be seen as an easily reproducible and strong baseline for future investigations of feature learning in exploration. Indeed, since the release of our paper, there has been some follow-ups on using random features for exploration in achieving state of the art results on hard exploration games when combined with extrinsic reward (in the interest of preserving anonymity, we don't include the references here).
>
> R2: "However, it isn't entirely clear if the primary contribution is showing that 'curiosity reward' is a potentially promising approach or if game environments aren't particularly good testbeds for practical RL algorithms"
> => We believe that both are valuable insofar as generating discussion within the community and leading to follow-up experimentation. In particular, we hope our paper stimulates both, an interest in trying out more realistic/stochastic environments, *and* further research on curiosity as a potential useful reward.  In addition to that, we have shown that curiosity could be a very strong baseline to compare against in future papers.  All these, we argue, are valuable to the progress and health of the field.
>
> R2: "Dealing with highly stochastic environments seems a potential fatal flaw of the assumptions of this method. However, as I said previously, this is probably a discussion worth having given the popularity and visibility of game-based testbeds"
> => We agree that significant amounts of stochasticity would break the method we used in the paper, and it is an important issue to be addressed by future work. Our vivid demonstration of this issue in the maze environment has already inspired some recent papers to look into, in particular, by incentivizing episodic reachability (in the interest of preserving anonymity, we don't include references to these, but we will include them in the final version of the paper).
>
> R2: "I think that more casual ML/RL researchers will find these results controversial and surprising while more experienced researchers will see curiosity-driven learning to be explainable primarily by the intuition..."
> R2: "Even the 'focused experiments' can be explained with the intuitive narrative that in the state/action space"
> => Indeed in our experience, although a few people were not surprised, most of them were very surprised at the agents being able to make progress without any any extrinsic rewards. This suggests that the game designers (similar to architects, urban planners, gardeners, etc.) are purposefully setting up curricula to guide agents through the task by curiosity alone [Lazzaro, 2004].
>
> R2: "consider [Mirowski et al., ICLR17] with respect to auxiliary loss + RL extrinsic rewards to improve performance"
> R2: "RL + auxiliary loss isn't evaluated in detail"
> => We will add a discussion of recent works that deal with navigation tasks in maze environments [Mirowski et. al. ICLR 2017, Jaderberg et. al. ICLR 2017] in the related works section. In contrast to these works, we don't assume privileged access to the maze environment in the form of depth estimation or loop closure supervision. Auxiliary tasks are an important component of RL and exploration methods, however, in this work we chose to focus on the most generic setting with minimal assumptions about the environment: providing raw observations in response to actions. In environments with privileged access we expect auxiliary tasks to benefit both curiosity-driven and extrinsic-reward-driven RL methods.

---

### Author Response · Authors · 2018-11-22
**Authors' Common Response to Reviewers**

We thank the reviewers for their insightful and helpful feedback. We are glad that the reviewers found the investigation carried out in the paper useful and systematic (R1), clear and well-written (R2,R3), well-contextualized with a high-quality presentation (R2). R2 says, "this is an interesting result that ostensibly will lead to a fair amount of discussion within the RL community (and already has based on earlier versions of this work)". All the reviewers have recommended acceptance of the paper.

We answer individual questions that the reviewers raised in the respective replies, and look forward to their follow-up advice.

---

### Meta-Review · Area_Chair1 · 2018-12-15

**Confidence:** 5
**Recommendation:** Accept (Poster)

**Metareview:**

The authors have extended previous publications on curiosity driven, intrinsically motivated RL with this broad empirical study on the effectiveness of the curiosity algorithm on many game environments, the merits of different feature sets, and limitations of the approach. The paper is well-written and should be of interest to the community. The experiments are well conceived and seem to validate the general effectiveness of curiosity. However, the paper does not actually have any novel contribution compared against prior work, and there are no great insights or takeaways from the empirical study. Therefore, the reviewers were somewhat divided on how confident they were that the paper should be accepted. Overall, the AC agrees that it is a valuable paper that should be accepted even though it does not deliver any algorithmic novelty.